# Resident Memory T Cells and Their Role within the Liver

**DOI:** 10.3390/ijms21228565

**Published:** 2020-11-13

**Authors:** Sonia Ghilas, Ana-Maria Valencia-Hernandez, Matthias H. Enders, William R. Heath, Daniel Fernandez-Ruiz

**Affiliations:** 1Department of Microbiology and Immunology, University of Melbourne, at the Peter Doherty Institute for Infection and Immunity, Melbourne, VIC 3000, Australia; avalencia@student.unimelb.edu.au (A.-M.V.-H.); menders@student.unimelb.edu.au (M.H.E.); wrheath@unimelb.edu.au (W.R.H.); danielfr@unimelb.edu.au (D.F.-R.); 2Australian Research Council Centre of Excellence in Advanced Molecular Imaging, University of Melbourne, Parkville, VIC 3010, Australia

**Keywords:** memory CD8^+^ T cells, resident memory T cells, liver

## Abstract

Immunological memory is fundamental to maintain immunity against re-invading pathogens. It is the basis for prolonged protection induced by vaccines and can be mediated by humoral or cellular responses—the latter largely mediated by T cells. Memory T cells belong to different subsets with specialized functions and distributions within the body. They can be broadly separated into circulating memory cells, which pace the entire body through the lymphatics and blood, and tissue-resident memory T (T_RM_) cells, which are constrained to peripheral tissues. Retained in the tissues where they form, T_RM_ cells provide a frontline defense against reinfection. Here, we review this population of cells with specific attention to the liver, where T_RM_ cells have been found to protect against infections, in particular those by *Plasmodium* species that cause malaria.

## 1. Introduction

The successful containment of infections relies on the speed with which immune responses of sufficient intensity are mounted. Immunological memory enables the long-term maintenance of a small fraction of those cells that responded to and resolved an earlier infection. The number of specific memory T cells generated after an infection, while declining over time, is generally larger than that of naïve T cells of the same specificities [1]. In addition, memory T cells display an enhanced antigen sensitivity, requiring lower levels of antigen for activation [2]. Memory T cells thus respond more rapidly and potently to pathogen invasion, and can exert efficient protection, potentially lifelong, against previously encountered infections. Different subsets of memory CD8^+^ T cells have been identified on the basis of their migratory properties, e.g., circulatory memory T cells and resident memory T cells (T_RM_ cells). The latter have recently emerged as important mediators of protection in peripheral organs, a common point of entrance of pathogens, by inducing rapid and local responses upon antigen recall [3]. By combining transcriptional and phenotypic features with different approaches to investigate residency, studies have identified T_RM_ cells in various disease models and within several tissue settings, including the liver. Importantly, strategies have been devised to favour the formation of T_RM_ cells through vaccination, achieving promising results, for example, in the case of herpes virus infection in the mucosa of the female genital tract [4] and *Plasmodium* infection of the liver [5,6,7,8,9].

The liver is essential for the maintenance of homeostasis and is central to many metabolic and immunological processes. Hepatic functions are tightly regulated; and disturbances that lead to liver diseases such as microbial infections, chronic inflammation or cancer can result in death. The liver is also the target of certain pathogens, such as *Plasmodium*, *Leishmania,* or *Listeria*, which infect and develop in this organ during stages of their life cycles. Given the highly protective capacity of memory T cells, and in particular of T_RM_ cells, studying the biology of these cells may aid the development of prophylactic and therapeutic strategies against life-threatening conditions associated with organ damage or infection. In this review, we will focus on recent advances in understanding memory T cell and T_RM_ cell biology, focusing on liver T_RM_ cells. Indeed, knowledge on this cell subset has been successfully implemented in the development of novel, highly effective immunization strategies against infectious diseases.

## 2. Memory T Cells

Shortly after activation, T cells generally differentiate into either short-lived effector cells (SLECs), expressing KLRG1, CX3CR1, and S1PR5, or memory precursor effector cells (MPECs), which are KLRG1^-^ CX3CR1^-^ and IL-7R^+^ [10,11,12,13]. T cell activation results in the formation of large numbers of SLECs, but these cells rapidly decline in numbers upon clearance of the infection. MPECs, however, are less numerous but become long lived memory cells and show a greater ability to generate recall responses [11]. IL-7R^hi^ cells comprise most of the memory cells at late time points (>8 months) after infection [14]. Importantly, this general classification is not exhaustive. Thus, while certain memory T cells (KLRG1^hi^, described in detail below) mainly arise from IL-7R^hi^ cells [11,12], a small proportion of IL-7R^lo^ cells can persist for prolonged periods [11], and display low expression of this marker in the spleen 60 days after lymphocytic choriomeningitis virus (LCMV) infection [12]. Indeed, splenic CX3CR1^-^ (KLRG1^lo^) effector T cells can give rise to all circulating and non-circulating memory T cell population, while CX3CR1^+^ (KLRG1^hi^) effector cells mainly differentiate into effector memory T cells after LCMV infection [13]. However, a peptide immunization model revealed that not all IL-7R^+^ cells in the spleen are long-lived [15] and, conversely, some KLRG1^hi^ T cells can persist for long periods of time, providing control against *Listeria* infection [16].

The establishment and long-term survival of MPECs and the memory T cells they give rise to, requires the cytokines IL-7 and IL-15 [11,17]. Downstream signaling after IL-15 and IL-7 recognition results in the expression of anti-apoptotic molecules, such as Bcl2 and Mcl1, shown to prevent the death of activated effector T cells and therefore to promote memory formation [18,19,20,21].

IL-15 signaling induces a metabolic switch from glycolysis, typical of effector T cells, to fatty acid oxidation [22], which comparably generates about 6 times more energy per unit of weight of substrate [23] and is essential to sustain memory T cell survival. Indeed, TRAF6 deficient T cells, presenting defective mitochondrial fatty acid oxidation, display an enhanced contraction phase after activation. In turn, stimulation of fatty acid metabolism in these cells with a drug that promotes AMP-activated kinases and circumvents the deficiency in TRAF6, prevents this decline in the number of activated T cells [24]. The expression of the chemokine receptor CCR7 on MPECs facilitates migration to T cell areas in secondary lymphoid organs along a CCL19 and CCL21 gradient. In these organs, T cells are exposed to IL-7 predominantly produced by stromal cells [25]. IL-7 is also produced by epithelial cells in organs such as the skin and the intestine [25]. As mentioned above, most memory T cells arise from the subpopulation of effector T cells that express IL-7R [11]; and IL-7 signaling has been linked with elevated fatty acid uptake and oxidation in CD8^+^ T cells through the induction of aquaporin 9 expression, a glycerol transporter that supports fatty acid uptake [26].

In the absence of IL-15, basal CD8^+^ T cell memory proliferation is impaired and leads to a progressive decline in memory T cell numbers [27,28]. In addition, under steady state conditions or after infection, mice lacking IL-15 display low numbers T_RM_ cells in liver and skin [29,30,31], suggesting this cytokine provides an important maintenance and/or developmental signal for resident memory T cells. However, more recent studies suggest that IL-15 dependency might not be absolute for CD8^+^ memory T cells or tissue-resident T cell populations in some organs, such as the mucosa and central nervous system, after viral infection [32,33,34].

Memory T cells were initially separated into two subsets based on the expression of the lymph node homing molecules CCR7 and CD62L, with CCR7^+^ CD62L^+^ cells being termed central memory (T_CM_), and CCR7^-^ CD62L^-^ cells, effector memory T (T_EM_) cells [35]. T_CM_ cells were found to migrate through lymphoid tissues, whereas T_EM_ cells were thought to traffic through peripheral tissues and the blood [35,36]. However, recent work has shown that T cell memory populations display a higher degree of complexity. Based on the expression of the chemokine receptor CX3CR1, CX3CR1^int^ peripheral memory (T_PM_) cells can be discriminated from CX3CR1^-^ T_CM_ and CX3CR1^hi^ T_EM_ cells [13]. Gerlach et al. showed that T_PM_ cells can also express CCR7 and CD62L, reflecting a T_CM_ phenotype. However, contradicting previous descriptions [35,36], this study found CX3CR1^int^ T_PM_ cells in tissues and the thoracic duct lymph, while CX3CR1^high^ T_EM_ cells were predominantly found in the blood. Gerlach et al. therefore concluded that T_PM_ cells and not T_EM_ cells embody the major migratory memory subsets in peripheral tissues [13]. Another memory T cell subpopulation described, in humans and mice, are termed memory T stem cells (T_SCM_) [37,38]. These cells are CD44^low^ CD62L^hi^, similarly to naive T cells, but can be further distinguished by the expression of Bcl2 and CD122 and, in mice, of Sca-1. Transcriptome analyses showed that T_SCM_ cells are the least differentiated memory subset population. T_SCM_, as their name suggests, can give rise to a variety of different T cell populations such as SLECs, T_EM_, and T_CM_ cells. Furthermore, the capacity of T_SCM_ cells for self-renewal, survival, and proliferation exceeds that of T_CM_ and T_EM_ cells. They are also of major interest in cancer research due to their superior anti-tumor response and resistance to chemotherapy [37,38,39].

## 3. Resident Memory T Cells

In addition to the aforementioned memory T cell subtypes, which all circulate throughout lymphoid and/or non-lymphoid organs, another subtype of memory T cells that reside in peripheral tissues, termed tissue-resident memory T (T_RM_) cells, became evident in the skin after infection with herpes simplex virus (HSV) type 1 [3]. These skin-resident CD8^+^ T cells were found to be in disequilibrium with circulating T cells, and efficiently controlled re-infection in a herpes simplex virus model [3]. T_RM_ cells have now been identified in virtually all organs in mice [40,41] and humans [42] including lymphoid and non-lymphoid tissues (Table 1). Recent evidence suggests that, upon restimulation, a small portion of these cells may seed back into the circulation [43,44]. However, the veracity of this conclusion is questioned by other studies that indicate T_RM_ cells remain localized to their niche even when exposed to antigen [45]. While we will focus on CD8^+^ T_RM_ cells in this review, T_RM_ cells can derive from both CD4^+^ or CD8^+^ T cells. T_RM_ cells have become a major focus of T cell research throughout the last decade as they are an essential first line of defense against pathogen invasion in most tissues.

### 3.1. T_RM_ Cell Development and General Features

Identification of cell surface markers that can clearly distinguish T_RM_ cells from other memory T cell subsets in both mouse and human tissues is complicated by the fact that no single marker associated with T_RM_ cells is exclusive to this cell subset. Different T_RM_ cell populations are known to share a common transcriptional signature [31]. However, they can adapt to their local microenvironment resulting in marker and cell feature variations from tissue to tissue [41]. Examples of this phenomenon will be given in the following paragraphs.

The cell surface molecule CD69 is a canonical marker of T_RM_ cells. This molecule promotes tissue retention by complexing with and antagonizing sphingosine-1 phosphate receptor 1 (S1PR1), a receptor that is required for tissue egress [46]. In mice, the majority, but not all of T_RM_ cells retained in tissues during parabiosis studies express CD69 [41]. In humans, sorting of CD69^+^ memory T cells from different tissues demonstrated a conserved transcriptional profile distinct from blood memory T cells and similar to that of mouse T_RM_ cells [47,48]. However, expression of CD69 is not sufficient to distinguish T_RM_ cells from other T cell subsets. One major issue is that T cells express CD69 upon TCR engagement, and hence local exposure to antigen may prevent distinction of T_RM_ cells from activated T cells. Exposure to type I IFN can also cause upregulation of this molecule on T cells [46], complicating T_RM_ cell identification during ongoing inflammation. Finally, CD69 has been shown to be dispensable for the generation and maintenance of T_RM_ cells in various tissues, such as liver, salivary gland, or lymph nodes [49]. Other markers are therefore necessary for T_RM_ cell identification.

Another marker widely used to identify T_RM_ cells is the molecule CD103 (the α subunit of the αEβ7 integrin), which binds E-cadherin expressed on epithelial and thus retains cells on the epithelium. This molecule is broadly expressed by murine T_RM_ cells from mucosa and barrier tissues [31,50,51,52]. However, murine T_RM_ cells from lymphoid organs and some non-barrier tissues such as the kidney and the liver do not express CD103 [6,53,54,55]. Similar observations have been made in humans where T_RM_ cells express CD103 in mucosa and barrier tissues but not in lymphoid organs [47,48,56,57]. Interestingly, unlike mice, some human liver T_RM_ cells do express CD103 [58]. This is thought to be related to the broad expression of E-cadherin by human hepatocytes [58], which may promote the retention of human T_RM_ cells within the liver. On the contrary, the retention of murine liver T_RM_ cells within the liver is achieved through the interaction of lymphocyte function-associated antigen-1 (LFA-1) with the intercellular adhesion molecule-1 (ICAM-1) expressed by the liver sinusoidal endothelial cells [55]. Thus, while CD69 and CD103 are useful markers to define T_RM_ cells from several tissues, they are not sufficient, and the context of expression must be considered when interpreting analyses.

More recent studies in both mice and humans have demonstrated the importance of the molecule CD49a in the biology of some T_RM_ cell subsets. This protein, also known as integrin α1, pairs with CD29 (integrin β1) to form the very late antigen (VLA-1), which binds to extracellular collagen and laminin and promotes the retention of T cells in tissues [59]. In peripheral tissues, like skin or liver, the majority, but not all, of murine and human T_RM_ cells express CD49a [3,48,60,61,62,63]. Importantly, in human skin, CD49a expression has been shown to discriminate two functionally different populations of T_RM_ cells, with CD49a^+^ T_RM_ cells producing IFN-γ and CD49a^-^ T_RM_ cells producing IL-17 [64]. CD49a may play a role in adhesion of T_RM_ cells to basement membranes of the epithelium. In support of this view, depletion of CD49a results in a decrease of memory T cells within the lung [59]. However, a recent study has shown that CD49a expression on T cells facilitates locomotion of virus specific CD8^+^ T cells in the trachea, suggesting that CD49a supports T_RM_ motility in this organ [63].

Other molecules have been identified as signature markers of T_RM_ cells in different tissues. For instance, the chemokine receptor CXCR6 is expressed by T_RM_ cells in several mouse organs like the liver and the lung, where it promotes respectively their maintenance and airway localization [6,65,66]. Likewise, human T_RM_ cells express CXCR6 across multiple tissues [48]. The molecules PD-1 and CD101 are also commonly expressed by T_RM_ cells from different tissues [45,48,67,68]. In contrast, most T_RM_ cells are negative for the chemokine receptor CX3CR1 [6], which is found on some circulating memory T cells in mice and humans [13,48]. Similarly, murine and human T_RM_ cells do not express KLRG1, nor lymph node homing molecules, such as CD62L, CCR7, or S1PR1 [40,48,69].

Environmental factors particular to each tissue, such as the expression of differential cytokines, can shape the formation and maintenance of T_RM_ cells. For example, tumor necrosis factor (TNF), IL-33, IL-15, IL-21, as well as transforming growth factor-β (TGF-β) have been shown to influence generation of T_RM_ cells in various non lymphoid tissues, such as the skin, salivary glands, or intestine [31,69,70,71]. As TGF-β is known to promote CD103 upregulation, and some T_RM_ cells such as those in the liver are CD103^-^, these cells are suggested to be maintained in a TGF-β independent manner. However, these cells are not unresponsive to TGF-β, as a recent RNA-seq based study revealed that TGF-β stimulation in vitro induced the upregulation of core signature T_RM_ cell genes in CD8^+^ T cells from several tissues, including the liver [72].

Transcription profiling has also highlighted a broad range of transcription factors associated with T_RM_ cell formation and/or maintenance. For instance, the development of several murine T_RM_ cell populations, including liver resident cells, requires cooperation of the transcription factors Hobit and Blimp1 [73]. Nonetheless, in humans, different observations have been made. For instance, while Pallett et al. found that human liver T_RM_ cells are Hobit^low^ Blimp1^high^ and suggested that Blimp1 compensates for the lack of Hobit upregulation [58], Stelma et al. showed that human liver T_RM_ cells express low levels of both molecules indicating that an alternative molecular mechanism could be involved in their differentiation process [74]. Indeed, it is possible that these studies looked at different subsets of T_RM_ cells: a recent study on memory CD8^+^ T cells in the murine intestine suggests that Blimp1 expression identifies functionally and transcriptionally distinct T_RM_ cell subsets [75]. Blimp1^high^ T_RM_ cells display strong effector capabilities and govern the early phase of acute infections whereas Blimp1^low^ T_RM_ cells are described as a memory population that persists long after infection [75].

### 3.2. Function of T_RM_ Cells

Upon re-exposure to a pathogen, T_RM_ cells provide a first line of adaptive cellular defense in peripheral non-lymphoid tissues. Mouse T_RM_ cells from various organs have been shown to mediate rapid protection against diverse bacterial, viral, and parasitic infections with more effective and rapid pathogen clearance compared with other subsets of memory T cells [3,6,53,85,91]. T_RM_ cells have also been associated with improved solid cancer prognosis (reviewed in [92]).

Upon antigen encounter T_RM_ cells rapidly produce different effector molecules including cytotoxic factors like granzyme B (GzmB) or perforin, and inflammatory cytokines such as Interferon-γ (IFN-γ) and Tumor Necrosis Factor (TNF) as observed in different organs and upon various infection model [6,93,94]. Hence, T_RM_ cells likely exert their protective function by either direct killing of infected cells or by attracting other immune cells to the site of infection. T_RM_ cells in the skin have been found able to clear HSV infection in the absence of circulating cells [95], and WT, but not IFN-γ or perforin-deficient T_RM_ cells in the brain were able to control intracerebral LCMV infection in mice depleted of circulating cells [94]. These findings suggest that T_RM_ cells can mediate direct killing of pathogens. Additionally, the chemokines and inflammatory cytokines produced by T_RM_ cells upon recall infection can trigger the recruitment and the activation of other inflammatory cells in particular circulating memory T cells [53,96,97]. As a consequence of their recruiting capacity, a small number of pathogen specific T_RM_ cells can trigger very rapid and efficient local immunity.

As a result of their remarkable protective capacities, T_RM_ cells have emerged as a promising means to combat infection and cancer. Indeed, recent studies on liver T_RM_ cells provide a clear example of the protective potential of these cells, as well as the opportunities to promote their formation through vaccination for effective immunity against infection.

## 4. Liver T_RM_ Cell Location

The liver is the recipient of both arterial and venous blood. The portal vein delivers large volumes of blood from the gastrointestinal tract and spleen to the liver [98]. Once there, the blood flows through narrow vascular capillaries known as hepatic sinusoids, which reduce the flow rate and allow resident cells to interact with a vast variety of antigens and circulating cells [99]. The hepatic sinusoids are lined with liver sinusoidal endothelial cells that form a fenestrated thin layer that separates hepatocytes from circulating cells. These fenestrae grant lymphocytes in the blood direct access to the surface of hepatocytes for antigen recognition and effector function [100,101]. In contrast to T_RM_ cells in most tissues, which are anatomically separated from the circulation, liver T_RM_ cells are present within the sinusoids and are constantly exposed to the blood stream but are able to access antigen on tissue stroma through the fenestrated endothelium [6]. Intravital images shows that liver T_RM_ cells, which display an ameboid shape, are uniquely located in the vasculature where they patrol the hepatic sinusoids at migration speeds more rapid than seen for skin T_RM_ cells (Figure 1) [6,41,73].

### 4.1. Identification of Liver T_RM_ Cells

Malaria is a major infectious disease caused by *Plasmodium* parasites. In their vertebrate host, parasites first develop in the liver for a short period of time, where they infect hepatocytes, before being released into the bloodstream to cause blood-stage infection, which leads to disease symptoms. Early evidence supporting the existence of resident memory T cells in the liver came from studies investigating the role of CD8^+^ T cells against the liver-stage of *Plasmodium*. These studies identified a long-lasting population of memory CD8^+^ T cells present in the liver and absent in the spleen of mice vaccinated with radiation-attenuated *Plasmodium* sporozoites (the infectious stage transmitted by the mosquito) [102]. Vaccinated mice were protected against *Plasmodium* sporozoite challenge for more than 6 months [102]. Later reports revealed that a subpopulation of memory CD8^+^ T cells associated with the liver, but absent from the circulation, expressed high levels of CXCR6, CXCR3, and CD69 [5,65], markers commonly displayed by T_RM_ cells [103].

The presence of bona fide memory cells permanently residing in the liver was confirmed by parabiosis studies in mice systemically infected with LCMV or *Plasmodium* sporozoites [6,41]. Parabiosis requires the surgical union of the flank skin of two animals. This enables the mixing of blood between the parabionts, and thus evaluation of T cell migration from one animal to the other. Unlike circulating cells, which equilibrate between both animals, resident populations remain in the parabiont in which they originally formed. This technique has been extensively used to identify T_RM_ cells in different murine tissues [41]. Although liver T_RM_ cells are in constant contact with circulating blood [6], parabiosis studies have confirmed that these cells, counterintuitively, do not recirculate and can only be found in the livers of the immunized parabiont partner [6,41].

Liver T_RM_ cells were found to express a similar phenotypic and transcriptional signature to that of T_RM_ cells previously identified in the lung, skin, and gut [6,31]. Maintenance of liver T_RM_ cells in mice relies on the expression of the transcription factor Hobit, and on basal levels of expression of Blimp1 [73]. These T_RM_ cell signatures have been found in T cells from grafted or isolated human tissues, enabling the unequivocal identification of T_RM_ cells in several human organs [48], including the liver [58,74]. As mentioned earlier, contrary to liver T_RM_ cells in mice which express high levels of Hobit and low to intermediate levels of Blimp1 [73], human liver T_RM_ cells are Hobit^low^ Blimp1^high^ [58]. In a recent publication, a small proportion of donor cells were found in HLA-mismatched liver and allografts 11 years after transplant, demonstrating the resident nature and remarkable longevity of these cells [104].

### 4.2. Liver T_RM_ Cell Immune Responses to Infection

Murine studies have shown that liver T_RM_ cells can confer efficient protection against liver-stage *Plasmodium* infection [6,9]. These studies have also demonstrated that substantial numbers of liver T_RM_ cells are associated with higher levels of immunity to malaria, and depletion of these cells ablates protection [6,9]. Based on these results, several complex vaccinations strategies, aimed at trapping activated CD8^+^ T cells in the liver, have now successfully induced the formation of liver T_RM_ cells in mice [6,7,8,9]. One vaccination strategy, prime-and-trap, is a single injection of a 3-component vaccine designed to prime *Plasmodium*-specific CD8^+^ T cells in the spleen and recruit them to the liver to form T_RM_ cells via locally expressed antigen recognition and adjuvant-induced inflammation [6,9]. Another strategy, termed prime and target requires the administration of two components injected two weeks apart and uses a modified adenovirus for priming and either nanoparticles or a modified viral vector to target cells to the liver [7]. More recently, we have also used a glycoprotein-peptide vaccination strategy that utilizes NKT cell help to induce the formation of liver T_RM_ cells [8]. In mice, vaccine-induced T_RM_ cells patrol the liver sinusoids, form aggregates around infected hepatocytes and, based on expression of molecules such as GzmB, IFN-γ and TNF-α (Figure 1) [6,7], potentially exert infection control through direct lysis and/or cytokine-mediated mechanisms. Moreover, vaccination studies with attenuated *Plasmodium* sporozoites in non-human primates have found high frequencies of intrahepatic memory CD8^+^ T cells in protected subjects [105].

Importantly, in humans, liver T_RM_ cells have been associated with disease control. For example, recent studies have investigated paired blood and liver samples from patients with chronic hepatitis B and hepatitis C virus infection and healthy volunteers to determine the role of liver T_RM_ cells during viral infections [58,74]. Researchers found that human T_RM_ cells in the liver express high levels of IL-2 and accumulate in larger numbers in the livers of infected patients compared to healthy patients. These studies also determined higher expression of GzmB and IFN-γ in HBV infected patients. Importantly, an inverse correlation between liver T_RM_ frequencies and viral titers was observed, indicating that high numbers of specific liver T_RM_ cells were associated with viral control [58]. However, accumulation of intrahepatic CD8^+^ CD103^+^ perforin^+^ T cells has been observed in cases of autoimmune hepatitis, particularly in indetermined pediatric acute liver failure [106]. These findings suggest that liver T_RM_ cells could also have a pathogenic function.

## 5. Conclusions

T_RM_ cells are pivotal mediators of protective immune responses within tissues and have been identified in nearly all organs, including lymphoid, non-lymphoid and barrier tissues. They are loaded with effector molecules, including GzmB, perforin, IFN-γ, and TNF, and likely exert their function by the direct killing of targets, or by recruiting other immune cells. Several infection models have correlated the presence of T_RM_ cells with pathogen and tumour control in tissues. Notably, in the liver, CD8^+^ T_RM_ cells can mediate efficient control of liver-stage Plasmodium parasites, and likely, HBV and HCV infections. For this reason, T_RM_ cells appear of particular interest in the course of vaccine development, especially for liver T_RM_ cells for malaria vaccines. Further research unveiling the mechanisms for the formation and maintenance of T_RM_ cells will facilitate the design of next generation T_RM_-based vaccines that realize the protective potential of these cells for unprecedented immunity against infections.

## Figures and Tables

**Figure 1 ijms-21-08565-f001:**
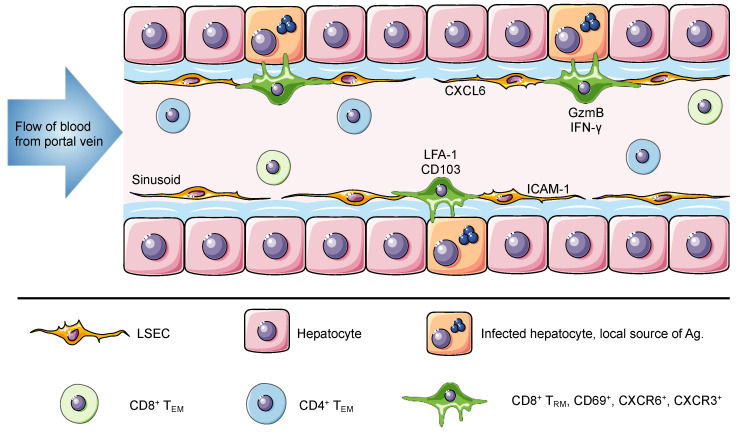
The liver is a unique niche for tissue resident memory cells. The portal vein delivers antigen-rich blood from the gastrointestinal tract and spleen to the liver. This blood flows through the liver hepatic sinusoids lined with a thin layer of fenestrated liver sinusoidal endothelial cell (LSEC). Liver T_RM_ cells are localized within the hepatic sinusoids, where they remain long-term and do not recirculate despite direct connection to the circulatory system and constant exposure to the blood. The expression of ICAM-1 and CXCL16 by LSEC can promote the retention of lymphocytes, through interactions with LFA-1 and CXCR6, respectively. Murine and human T_RM_ cells in the liver express CD69, CXCR6, CXCR3 and high levels of LFA-1. Of note, human but not murine T_RM_ cells express CD103. It has been suggested that this difference is associated with a broad versus a restricted expression of E-cadherin by human and murine hepatocytes, respectively. Intrahepatic lymphocytes including circulating and resident memory cells can access the surface of hepatocytes through LSEC fenestrae and exert effector functions. Using cytoplasmic protrusions, lymphocytes probe hepatocytes for the presence of antigen and can release factors such as GzmB and IFN-γ to promote hepatocyte killing. In murine studies, liver T_RM_ cells can be generated through different vaccination strategies to confer protection against *Plasmodium* parasites and in humans they have been associated with disease control against HBV and HCV.

**Table 1 ijms-21-08565-t001:** Expression of the canonical markers used to define CD8^+^ T_RM_ cells in diverse murine and human organs.

Organs	Expression of Canonical Markers (CD69, CD103, CD49a and CXCR6)			
	Mice		Humans	
**Intestine, Gut**	CD69+ CD103+/− CD49a+ CXCR6+	[40,41,52,76,77]	CD69+ CD103+	[64,78]
**Skin**	CD69+ CD103+/− CD49a+ CXCR6+	[31,79]	CD69+ CD103+/− CD49a+/−	[64,80]
**Lungs**	CD69+ CD103+ CD49a+ CXCR6+/-	[59,66,81]	CD69+ CD103+ CD49a+ CXCR6+	[47,48]
**Female reproductive tract**	CD69+/− CD103+/−	[40,41,82]	CD69+ CD103+ (transcriptomic profiling is yet to be determined)	[83,84]
**Salivary glands**	CD69+/− CD103+/− CD49a+	[41,85]	CD69+ CD103+/−	[48]
**Lymphoid organs (Spleen, lymph nodes, tonsil)**	CD69+ CD103− CD49a+	[53,86]	CD69+ CD103+/− CD49a-	[87]
**Liver**	CD69+ CD103− CD49a+ CXCR6+	[6,62,73]	CD69+ CD103+/− CXCR6+	[58,74]
**Kidneys**	CD69+/− CD103-	[40,41,54]	CD69+ CD103+/− CD49a+/− CXCR6+/−	[88]
**Pancreas**	CD69+/− CD103+/−	[40,41]	CD69+ CD103+ CD49a+ CXCR6+	[67]
**Brain**	CD69+ CD103+/−	[40,68,89,90]	CD69+ CD103+/− CD49a+ CXCR6+/−	[61]

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
