# Peer review of "Resident Memory T Cells and Their Role within the Liver"

_ijms, 2020, doi:10.3390/ijms21228565_

Round 1

Reviewer 1 Report

This review by Ghilas et al discusses novel insights into tissue resident memory CD8 T cell with primary focus on the liver . This is a clinically relevant topic, given evidence in their protective role against liver infections as well as tumors, and thus optimizing vaccine strategies to preferentially induce this population. The manuscript highlights some complexities in better defining these populations phenotypically as well as functionally and discusses both results in rodent  models as well as humans. The topic and content areas are overall presented and summarized in a Figure.

This reviewer has minor concerns and suggestions for improvement:

  1. Section line 57-69: The text summarizes some general phenotypic characteristics of SLEC and MPEC, but also mentions plasticity. It would help if the authors clarified some differences in the systems discussed. e.g. tissue, microenvironment, or pathogen dependence, and whether the statements refer to circulating or resident T cells. (This is touched upon in section 3.1, but should get more attention in he earlier section.)
  1. line 90-96. The authors do not discuss CD8 T cells in the CNS, which may also be IL-15 independent (Zuo J et al.. J Neuroimmunol. 2009 207(1-2):32-8; doi: 10.1016/j.jneuroim.2008.11.005)
  2. Section 4: Figure 1 and the narrative indicate liver resident TRM largely locate to the hepatic sinusoids, which are an extension of blood vessels. Can the authors clarify if these CD8 T cells are trapped transiently or reside locally more long term? Are CD8T cells also found directly in parenchymal tissue? The parabiosis data and transplant data would suggest they are not replaced by circulating T cells, which appears to conflict with their sinusoid location? These issues should be discussed and perhaps included in the Figure, even if controversial or not clear.
  3. There are 2 issues not discussed which may be of additional interest to the reader and should be touched upon: Is there any indication that the liver TRM may be pathogenic under certain conditions? Do the CD8 TRM require local CD4 T cell help.

Author Response

Thank you for your decision on our manuscript entitled ‘Resident memory T cells and their role within the liver’ (ijms-975630). We greatly appreciate your constructive comments. We have now addressed these comments, and we believed this has strengthened the review. You will find the revisions in red and highlighted as comments in the new version of the manuscript.

We outline point by point responses to your comments. We hope our revised version will be received favourably and look forward to hearing from you in the near future.

Point 1: Section line 57-69: The text summarizes some general phenotypic characteristics of SLEC and MPEC, but also mentions plasticity. It would help if the authors clarified some differences in the systems discussed. e.g. tissue, microenvironment, or pathogen dependence, and whether the statements refer to circulating or resident T cells. (This is touched upon in section 3.1 but should get more attention in the earlier section.)

Response 1: We added in this section more details about where and how this plasticity has been observed. (Line 65-73)

Point 2: line 90-96. The authors do not discuss CD8 T cells in the CNS, which may also be IL-15 independent (Zuo J et al.. J Neuroimmunol. 2009 207(1-2):32-8; doi: 10.1016/j.jneuroim.2008.11.005)

Response 2: We apologise for omitting that CD8+ memory T cells in the central nervous system are also IL-15 independent. We added this information as well as the related paper in the text. (Line 94-100).

Point 3: Section 4: Figure 1 and the narrative indicate liver resident TRM largely locate to the hepatic sinusoids, which are an extension of blood vessels. Can the authors clarify if these CD8 T cells are trapped transiently or reside locally more long term? Are CD8T cells also found directly in parenchymal tissue? The parabiosis data and transplant data would suggest they are not replaced by circulating T cells, which appears to conflict with their sinusoid location? These issues should be discussed and perhaps included in the Figure, even if controversial or not clear.

Response 3: Thank you for this comment. We tried to clarify our statements along the review and are hoping this is clearer. (Line 255-257, line 264-265 and line 297-299)

Indeed, parabiosis and transplant data suggest that liver TRM cells are not replaced by circulating T cells: in mice, liver TRM cells of vaccinated mice display a half-life time of 76 days (https://doi.org/10.1016/j.chom.2020.04.010) and in human, donor TRM cells are found 11 years after transplant in the liver of recipient patients (https://doi.org/10.1084/jem.20200050). But intravital live imaging (http://dx.doi.org/10.1016/j.immuni.2016.08.011) and intravascular antibody injection (http://dx.doi.org/10.1016/j.cell.2015.03.031) experiments have shown that in mice liver TRM cells are present within the blood vessels (sinusoids) and exposed to the circulation.

Point 4: There are 2 issues not discussed which may be of additional interest to the reader and should be touched upon: Is there any indication that the liver TRM may be pathogenic under certain conditions? Do the CD8 TRM require local CD4 T cell help.

Response 4: This is a fair comment. We included in the review a study showing that in autoimmune hepatitis, CD8+ CD103+ perforin+ T cells accumulate in the liver (https://doi.org/10.1002/hep.29901) suggesting that liver TRM cells could have pathogenic function. To our knowledge, this is the only paper suggesting a potential pathogenic role for liver TRM cells. (Line 339-341)

As far as we are aware, there is no study discussing the potential liver TRM cell requirement for CD4+T cell help. Thus, we didn’t discuss that in the review. However, in the lung, it has been showed that IFN-g produced by CD4+T cells was necessary for the generation of CD103+ CD8+ TRM cells during influenza infection (http://dx.doi.org/10.1016/j.immuni.2014.09.007). Moreover, a very recent study showed that CD4+T cell derived IL-21 can drive brain TRM cells differentiation during viral infection (DOI: 10.1126/sciimmunol.abb5590).

Thank you for your consideration

Sincerely,

Sonia Ghilas, PhD

Reviewer 2 Report

In this review article, Ghilas et al. provide a solid overview of CD8+ TRM cells, albeit one where a sizeable number of recent reviews are already available. Despite the title and text stating that their review focuses on liver-CD8+ TRM cells, this topic is only covered in section 4. In fact, it’s done so expeditiously as to make this reviewer come away feeling somewhat shortchanged, especially given the excellent papers on liver-CD8+ TRM cells and protection from malaria published by Dr. Fernandez-Ruiz and colleagues. What would help this review stand out from the many other TRM reviews is for it to delve more deeply into the biology of liver-resident TRM cells.

Specific Comments”

The authors should also discuss recent papers from the Masopust lab and others showing that TRM are capable of retro-trafficking from tissues to draining lymph nodes and to converting to circulatory TCM (10.1016/j.immuni.2018.01.015; 10.1084/jem.20192197).

In Table 1, the authors should include references from the J. Lokensgard and A. Lukacher groups on the expression of markers for TRM cells in the mouse brain.. Also, the title for this table should qualify the TRM as being CD8+.

Line 30:  change “overtime” to “over time”.

Lines 38-39: correct the dangling participle.

Line 44: insert a paragraph return.

Line 83: change “into” to “in”

Line 95: change “tissues-resident” to “tissue-resident”

Line 106: include citations for “contracting previous descriptions”

Line 114: change “TSCM cells…” to “the capacity of TSCM for self-renewal, survival and proliferation capacity exceeds …”

Line 152: technically CD103 is the alphaE subunit of the integrin heterodimer with beta7.

Line 161: spell out abbreviation “ICAM-1”

Line 181: include additional citations for PD-1 expression by CD8+ TRM cells (e.g., 10.1016/j.celrep.2019.11.056; 10.1038/icb.2017.69)

Line 188: capitalize the "T" in “Tumor Necrosis Factor”

Line 189: IL-21 should be added to the list of TRM cell inducing cytokines (e.g., 10.4049/jimmunol.1401236; 10.1126/sciimmunol.abb5590)

Line 254 (Fig 1 legend): Change “CXCL16” to “CXCL6”

Lines 289 vs 292: Liver TRM cell maintenance depends on Hobit (line 289), but express low levels of Hobit (line 292). Please resolve this apparent discrepancy.

Line 302: change “3-components” to “3-component”

Line 322: change “are associated” to “were associated”.

Author Response

Thank you for your decision on our manuscript entitled ‘Resident memory T cells and their role within the liver’ (ijms-975630). We greatly appreciate your constructive comments. We have now addressed these comments, and we believed this has strengthened the review. You will find the revisions in red and highlighted as comments in the new version of the manuscript.

We outline point by point responses to your comments. We hope our revised version will be received favourably and look forward to hearing from you in the near future.

Point 1: The authors should also discuss recent papers from the Masopust lab and others showing that TRMare capable of retro-trafficking from tissues to draining lymph nodes and to converting to circulatory TCM (10.1016/j.immuni.2018.01.015; 10.1084/jem.20192197).

Response 1: Thank you for this comment, we initially had a paragraph discussing this TRM cell feature, but we removed it due to lack of space. We agree that these studies are important in the field, and we included back a brief discussion of this TRM cell feature (Line 130-133).

Point 2: In Table 1, the authors should include references from the J. Lokensgard and A. Lukacher groups on the expression of markers for TRM cells in the mouse brain. Also, the title for this table should qualify the TRM as being CD8+.

Response 2: We included the references and modified the title of the table as you suggested. (Line 219)

Point 3: Line 30:  change “overtime” to “over time”.

Response 3: Corrected. (Line 30)

Point 4: Lines 38-39: correct the dangling participle.

Response 4: We modified the sentence as follows: By combining transcriptional and phenotypic features with different approaches to investigate residency, studies have identified TRM cells in various disease models and within several tissue settings, including the liver. (Line 38-40)

Point 5: Line 44: insert a paragraph return.

Response 5: Done (Line 44)

Point 6: Line 83: change “into” to “in”

Response 6: Corrected. (Line 87)

Point 7: Line 95: change “tissues-resident” to “tissue-resident”

Response 7: We modified the sentence and corrected this mistake. (Line 99)

Point 8: Line 106: include citations for “contracting previous descriptions”

Response 8: We included the references. (Line 110)

Point 9: Line 114: change “TSCMcells…” to “the capacity of TSCM for self-renewal, survival and proliferation capacity exceeds …”

Response 9: The sentence has been changed as suggested. (Line 118)

Point 10: Line 152: technically CD103 is the alphaE subunit of the integrin heterodimer with beta7.

Response 10: This is a good point. We clarified it. (Line 160-161)

Point 11: Line 161: spell out abbreviation “ICAM-1”

Response 11: Done. (Line 170 and 365 in the abbreviation section)

Point 12: Line 181: include additional citations for PD-1 expression by CD8+ TRM cells (e.g., 10.1016/j.celrep.2019.11.056;1038/icb.2017.69)

Response 12: We included the research article as a reference, as well as another one showing PD-1 expression in brain TRM cells. (Line 191)

Point 13: Line 188: capitalize the "T" in “Tumor Necrosis Factor”

Response 13: Done. (Line 197)

Point 14: Line 189: IL-21 should be added to the list of TRMcell inducing cytokines (e.g., 10.4049/jimmunol.1401236; 10.1126/sciimmunol.abb5590)

Response 14: This is a fair suggestion. We added IL-21 as an important factor for CD8+ TRM cells differentiation. (Line 198)

Point 15: Line 254 (Fig 1 legend): Change “CXCL16” to “CXCL6”

Response 15: We acknowledge your comment, but here we actually meant CXCL16, which is the ligand for CXCR6. (Line 266)

Point 16: Lines 289 vs 292: Liver TRMcell maintenance depends on Hobit (line 289), but express low levels of Hobit (line 292). Please resolve this apparent discrepancy.

Response 16: This seems to be a discrepancy, but it was maybe not well explained. Here we meant to say that although human TRM cells share signatures with mice TRM cells, these cells are Hobithigh in mouse liver but Hobitlow in human liver. (Line 306-307)

Point 17: Line 302: change “3-components” to “3-component”

Response 17: Corrected. (Line 318)

Point 18: Line 322: change “are associated” to “were associated”.

Response 18: Corrected. (Line 338)

Thank you for your consideration

Sincerely,

Sonia Ghilas, PhD